# Effect of Beam Oscillation on Microstructure and Mechanical Properties of Electron Beam Welded EN25 Steel

**DOI:** 10.3390/ma16072717

**Published:** 2023-03-29

**Authors:** Vasundhara Singh, Prakash Srirangam, Gour Gopal Roy

**Affiliations:** 1Department of Metallurgical and Materials Engineering, IIT Kharagpur, Kharagpur 721302, India; vasuhappy@iitkgp.ac.in (V.S.); ggroy@metal.iitkgp.ac.in (G.G.R.); 2Warwick Manufacturing Group, University of Warwick, Coventry CV4 7AL, UK

**Keywords:** beam oscillation, electron beam welding, EN 25 steel, hardness, mechanical properties, nonmetallic inclusions, residual stress, wear

## Abstract

EN25 steels have been found to be applicable in shafts, gears, etc., but welding of EN25 steel was performed using electron beam welding with different oscillation beam diameters varying from 2 mm to 0.5 mm. The present study reports the effect of beam oscillation on the evolution of nonmetallic inclusions, microstructures, and mechanical properties of EN25 steel. Heat input calculations showed that the application of beam oscillations resulted in significantly lower heat inputs compared to their non-oscillating counterparts. The highest fraction of the retained austenite (9.35%) was observed in a weld prepared with beam oscillation at a 2-mm oscillation diameter, and it decreased to 3.27% at an oscillating diameter of 0.5 mm, and it further reduced to 0.36% for non-oscillating beam cases. Residual stresses were compressive in the fusion zone, irrespective of beam oscillation. Beam oscillation resulted in equiaxed grain in the recenter region of the fusion zone, attributed to heat mixing and the evolution of random texture. The application of beam oscillations resulted in a significant decrease in the size of the nonmetallic inclusions to 0.1–0.5 compared to 5–20 mm in base metal. All tensile samples failed in the base metal, indicating good strength of the weld. Fusion zone hardness (250–670 HNV) and wear properties (COF 0.7 to COF 0.45) improved irrespective of with and without beam oscillation.

## 1. Introduction

Medium carbon low alloy (MCLA) Ni–Cr–Mo steels have been a center of attraction for many researchers in the last few decades. Several studies have been conducted on MCLA steels due to their excellent balance of strength, toughness, wear resistance and weldability. European-grade (EN) Ni–Cr–Mo steels find application in machine part members, gears, and shafts due to their high hardenability, good tensile strength and toughness, which could be tailored by proper heat treatment [1]. EN25 is used to make gears, motor shafts, axle shafts, connecting rods, torsion bars, adapters, spindles, die holders, piston rods, oil refining, high-temperature bolts and steam installations [2].

Welding is the technological reconditioning process that ensures short repair time. Electron beam welding (EBW) is a fusion welding process in which a high-energy density beam (10^7^ W/cm^2^) is used to join the metal parts, which results in the formation of a contamination-free narrow heat-affected zone (HAZ) with low distortion compared to conventional welding techniques [3,4,5,6,7,8,9]. EBW has been used for joining high-strength steel in aerospace applications and several studies have been conducted to understand the effect of EBW parameters on the microstructure and properties of HSLA steel components [10].

Arata et al. [11] investigated the behavior of the EB weld zone in low-alloy steels (Ni–Cr–Mo steels). In the weld zone, four types of cracks were found, namely horizontal, vertical crack I, vertical crack II and cold shut. Cracks in the weld zone were mainly due to solidification except for vertical crack II, which was formed to occur along the prior austenite grain boundary due to high hardenability in that region.

Ueyama et al. [12] worked on Ni–Cr–Mo steel to understand the crack behavior and found that with increasing laser power, both the penetration and crack length in solidification cracking increased, suggesting a strong correlation between crack length and penetration. Qiang Wang et al. [13,14] proposed a new effective stress intensity factor range parameter as the crack growth driving force for Ni-Cr-Mo-V high-strength steel, which yielded a fairly good correlation for the mixed mode fatigue crack growth data. Acicular ferrite with a basket-weave microstructure in the weld metal exhibited favorable crack growth resistance relative to the base material. The damage tolerance design discussed this HSLA steel in a marine environment. Shi-Dong Liu et al. [15] showed that short fatigue crack growth was influenced by both a local microstructure and a global strength gradient. It is indicated that strain localization at lathy boundaries and the formation of sub-grains were responsible for the fatigue of short cracks.

Sadeq et al. [16] studied the discontinuous welding repair of worn carbon steel shafts through arc welding. The repaired area contained a large region of soft ferrite and discontinuous regions of pearlite, which increased hardness and wear resistance while preserving the tension strength as it was before the repair.

Sisodia et al. [17] studied the microstructural and mechanical properties of S960M high-strength steel by EBW. Base material (BM) contained upper bainite and autotempered martensite, while the fusion zone (FZ) consisted primarily of martensite and the heat-affected zone (HAZ) comprised martensite and bainite with a small amount of ferrite. The highest average hardness was observed in the fine-grained heat-affected zone (FGHAZ); HAZ was harder than FZ. The tensile strength of the EB-welded joint (997 MPa) reached the level of BM (1058 MPa). Lateral expansion showed HAZ (1.99 mm) and FZ (0.59 mm) had brittle-ductile characteristics and lower expansion compared to BM (1.34 mm). Charpy fractures of BM exhibited ductile failure, while HAZ and FZ showed brittle-ductile characteristics.

Bai et al. [18] studied the weldability of SA508Gr4 steel for nuclear pressure vessels. SA508Gr4 steel exhibited a high hardening tendency and high cold-cracking susceptibility during welding. The results showed that when the HAZ cooling time was less than 15 s, lath-shaped martensite developed, which resulted in extensive hardening and cold cracking in the HAZ. Cooling time above 1200 s resulted in bainite formation, which suppressed cracking. Hence it was suggested that preheating to 196 °C or higher improved the weld quality. 

Kar et al. [19] and Dinda et al. [20] studied the effects of beam oscillation on the microstructures and mechanical properties of dissimilar EB welded joints between SS-Cu and steel and Fe–Al alloy, respectively. They demonstrated that the ductility of the joint increased significantly compared to its non-oscillating counterpart, which was attributed to heat mixing and the development of more uniform, equiaxed microstructures and random textures [19,20,21,22].

Nayak et al. [21] investigated the role of beam oscillation on electron beam welded zircaloy-4 butt joints and reported that the application of beam oscillation resulted in a more uniform and fine-basket weave widmanstätten microstructure in the fusion zone of the joints. Wang and Wu [23] reported that a linear oscillating beam is better than a circular one because the former promotes the refinement of the fusion zone microstructure, while later producing a coarser microstructure. Wang et al. [24] reported that joints produced with a higher welding speed possessed comparatively fine-grained structures with higher tensile strength. Doong et al. [25] reported that beam oscillation significantly reduced the porosity content and improved the fatigue life of the 4130 steel joints. Xia et al. [26] reported that beam oscillation induced the early formation of equiaxed grain at fusion zone depth direction and improved the weld morphology and uniformity of microstructures. Several studies have been conducted on the evolution nonmetallic inclusions (NMIs) with weldability of steels, which has been found helpful towards property enhancement. Jincheng Sun et al. [27] studied the effects of heat input on inclusion evolution behavior in the heat-affected zone of EH36 shipping steel. This was performed systematically through ex-situ (scanning electron microscopy) SEM examination and in-situ (confocal laser scanning microscopy) CLSM observation. Al-Mg-Ti-O-Mn-S is a type of complex inclusion as observed in such steel. The count of inclusions (number density) remained constant, but the MnS number density decreased with increasing heat input. Low heat input produced nucleation of acicular ferrite on inclusions, which increased the toughness of the HAZ. Chen et al. [28] and X. Wan et al. [29] reported that in HSLA, TiN is a stable inclusion at high temperatures, which retards the austenite grain growth by refining the grain size. The crystallographic grain size became small in the simulated HAZ due to the effective pinning effect and acicular ferrite formation.

It is evident from previous research studies that a lot has been studied on the effect of weld parameters on the microstructure and mechanical properties of steels. However, in this study, we used the EBW technique with different beam oscillation diameters to understand the changes in inclusions and microstructure formation in EN steel grade, which has not been extensively studied in the past. 

## 2. Materials and Methods

### 2.1. The Investigated Base Material

Two Ni-Cr-Mo steel plates (EN25 steel) were used as the base material for electron beam welding of similar joints. EN25 steel was provided by Heavy Engineering Corporation Limited Ranchi, India. Chemical analysis of base material used in joining was analyzed using X-ray spectroscopy analysis is presented in Table 1. To perform welding, the base material was cut into 32 mm × 26 mm × 3 mm (Length × Breadth × Height) dimensions using a diamond wheel cutter with a water-cooling system. The sample was polished using 220, 600 and 1200 grit then, it was given ultrasonic cleaning and finally, it was cleaned with acetone. 

### 2.2. Experimental Procedure

The worksheet for the current study is presented in Figure 1a. Electron beam (EB) welding was performed in a butt-welding configuration with and without beam oscillation (as shown in Figure 1b). An electron beam weld (EBW) on an EN25 steel bead on a plate was performed on three samples using the EBW machine at the Bhabha Atomic Research Centre (BARC), Mumbai, India. Table 2 presents a list of the operating parameters for EBW. For electron beam welding, the process variables used were gun chamber vacuum (mbar) = 10^−6^, welding chamber vacuum (mbar) = 10^−5^, both base material dimensions = 32 × 26 × 3 mm^3^ and gun specimen distance (mm) = 465. The oscillation beam shape was circular. 

Microstructural studies were carried out using an optical microscope and scanning electron microscopy (SEM) on the welded samples. Samples for microscopy were mechanically polished and etched using 2% Nital (2 mL HNO_3_ and 98 mL C_2_H_5_OH) and hot picric acid etchant (2% picric acid and distil water 100 mL and HCL 2–3 drop and soap solution). Grain size measurements were carried out using the grain intercept method. SEM analysis was carried out using Zeiss (fitted in Zeiss^®^ EVO 60 SEM) from Oxford Instruments (Oxford, UK).

Electron backscattered diffraction (EBSD) studies were performed at a step size of 0.2 μm using the TSL OIM analysis software (fitted in Zeiss^®^ Auriga compact dual beam scanning electron microscope) from Oxford Instruments, UK, operated at 20 kV over the sample before and after EB welding. Sample surfaces were gently polished with an aqueous colloidal silica solution before EBSD scanning.

Microhardness measurements were performed using a Wilson hardness testing machine. Hardness measurements were taken along the transverse direction of the weld bead, heat affected zone (HAZ) and the fusion zone (FZ) using a diamond pyramid indenter under a load of 100 g with a dwell time of 15 s. Ten indentations per region were performed, which were then averaged to get the overall Vickers Hardness (HV) in each case with a standard deviation ±5.

Tensile tests were performed using an Instron tensile testing machine with a 10 kN maximum load capacity fitted with a digital extensometer. Figure 2 represents the schematic of the ASTM E-8 subsize standard tensile specimen geometry. A strain rate of 0.2 mm/min was used in tensile tests for all five weld samples as well as for base metals. For each weld condition, tests were repeated three times to report the average values with a standard deviation ±5.

X-ray diffraction experiment was performed in Bruker D8 discover diffractometer for quantification of retained austenite (RA) and residual stress calculation for welds with different welding parameters. Adequate polishing and ultrasonic cleaning were performed on the samples to avoid any contamination on the surface from the grinding. The Co-Kα radiation (wavelength: 1.789 Ả) was selected as the incident X-ray. The strongest reflections (111), (200) from austenite and (110), (200), and (211) from ferrite were used for the estimation of retained austenite from the Rietveld refinement method using TOPAS software. Lattice strain, crystallite size, dislocation density and residual stresses were also calculated from the XRD data plot.

The kinetics of wear were studied using a fretting wear testing unit. The wear test was performed using a ball-on-disk wear tester (Ducom-TR-283M, Ducom, Bohemia, NY, USA). The wear test was conducted at a constant load of 30 N for a constant testing duration of 30 min at a constant frequency of 10 Hz and a constant stroke length of 1 mm. The wear data were analyzed using Winducom2006 software. The variation of wear depth with time was studied. The microstructure of the worn-out debris was analyzed with the SEM to understand the mechanism of wear. Before carrying out the test, all the samples were diamond polished and cleaned properly. For each weld condition, tests were repeated three times to report the average values with a standard deviation ±5.

## 3. Results and Discussion

### 3.1. Microstructural Analysis

#### 3.1.1. Base Material (BM)

Figure 3a,b represents the base material microstructure after polishing and etching under optical and SEM micrographs. The base material consists of martensitic plates and martensitic blocks formed around a prior austenite grain boundary as this material was forged and quenched. No intermetallic compounds were observed in EN25 steel.

#### 3.1.2. Macroscopic Images after Welding with Base Metal (BM), Heat Affected Zone (HAZ) and Fusion Zone (FZ)

Figure 4 shows the cross-sectional appearance of weld bead under four different welding conditions. Figure 4a represents the weld with an oscillation diameter of 2 mm. It shows a lack of fusion at the top, attributed to low heat input with a higher oscillation diameter. The weld obtained had a 0.5-mm oscillation diameter and showed a better appearance of the fusion zone top. Similarly, with an increase in speed under without beam oscillation conditions, incomplete penetration at the top occurs at a higher speed (Figure 4d), corresponding to a low heat input. 

Figure 5 represents the optical microscopy images of the fusion zone with and without beam oscillation conditions in electron beam welding. It is observed that the microstructures are predominantly columnar, except in two cases where equiaxed grains are seen significantly in the weld zone, namely, beam oscillation with a 2-mm oscillation diameter and weld without beam oscillation at maximum weld speed. Equiaxed grains may be promoted either at a high cooling or solidification rate, where a low G/R promotes equiaxed dendritic structure or under beam oscillation, when the heat mixing promotes crystal growth in multi-directions [19,20,21,30].

#### 3.1.3. Heat Input Calculation for Different Welding Conditions

Beam oscillation changes the heat input rate to the material and consequently affects the evolution of the microstructure. Therefore, this section presents the heat input calculation and correlates it with the evolution of microstructure, inclusion and properties. Heat input per unit length is calculated during welding using Equation (1) given below:(1)$$Heat\;Input\;rate\;\left(Q\right)\;=\;\eta \frac{V*I}{v}$$
where ***Q*** = heat input rate during the welding operation in kJ/mm, ***η*** = efficiency of power supply usually 0.9 for EBW, ***V*** = voltage used during the welding operation, ***V***, ***I*** = current used during the welding operation, mA and ***v*** = speed of the beam, mm/min.

During non-oscillating conditions, the speed of the beam is equivalent to the scan speed of welding. During oscillating electron beam welding, the velocity of the electron beam changes and can reach values of several thousand mm/s, depending on the welding velocity and the oscillation parameters (oscillation diameter and frequency).

In the case of the oscillating beam, the velocity of the beam is calculated as shown in Equation (2), as reported in the literature [31]:(2)$$v\;=\;\sqrt{v_{x}{}^{2}+v_{y}{\;}^{2}}$$
(3)$$v_{x}\;=\;\frac{dx\left(t\right)}{\;dt\;}=2.\;𝜋.\;f_{x}.\;a_{x}.\cos\left(2\pi \;f_{x}t+\Phi_{x}\right)$$
(4)$$v_{y}\;=\;\frac{dy\left(t\right)}{\;dt\;}\;=\;v_{w}+2.\;𝜋.\;f_{y}.\;a_{y}.\cos\left(2\pi \;f_{y}t+\Phi_{y}\right)$$
where ***a*** and ***f*** represent the oscillating amplitude and frequency, respectively. ***ϕ*** represents the initial phase angle. Subscripts **x** and **y** represent the x- and y-component, respectively. For circular beam oscillation patterns, the amplitude and frequency are equal, and the phase shift is π/2. $v_{w}$ is the welding scan velocity. By using Equation (2), the velocity of the beam is calculated at different beam oscillation diameters. After calculating the heat input using Equation (1), the value of the heat input with a beam oscillation diameter of 2 mm was found to be the lowest of all the welding parameters taken, as presented in Table 3. Weld beads without beam oscillation were the lowest at the highest speed when compared to other speeds. Similar results were reported by S. Dinda for EBW dissimilar steel to Fe-Al alloy joints [20]. Table 3 also shows the weld bead dimensions. It is found that weld bead thickness increases with an increase in heat input. Consequently, a weld bead with beam oscillation at a 2-mm oscillation diameter shows the minimum weld bead thickness, which is highest in cases without beam oscillation and at the lowest speed. 

#### 3.1.4. EBSD Analysis of the Welded Region

Figure 6 represents the IPF map with and without beam oscillation conditions. The finer grain size was confirmed in the welded region with a beam oscillation diameter of 2 mm, as seen in Figure 6a. Grains became coarser for the beam oscillation diameter of 0.5 mm and a without beam oscillation condition, maintaining the speed at 1000 mm/min, voltage at 60 kV and current at 35 mA for all welding conditions. The base metal grain size was found to be the highest, as seen in Figure 6d. The solidification structure became finer with the increased cooling rate and the cooling rate increased from the sample without oscillation to samples prepared with beam oscillation at the highest oscillation diameter (Table 3). Therefore, the finest grains were obtained with beam oscillation at a 2-mm oscillation diameter and the coarsest samples were produced in the option without beam oscillation. The base metal is a high-temperature solution-forged sample that shows coarse grains because prior austenite gets coarser at high temperatures.

The grain size distribution is presented in Figure 7 for all sample conditions. It shows that grain refinement takes place under the condition of beam oscillation, and it is lowest at the highest beam oscillation diameter, which may be attributed to the lowest heat input. However, grain coarsening was observed for withoutbeam oscillation case weld counterparts, especially with low weld scan speed. 

### 3.2. Residual Stress, Lattice Strain and Dislocation Density from XRD Analysis

The X-ray diffraction pattern of the EN 25 steel fusion zone with beam oscillation and without beam oscillation is shown in Figure 8. The XRD reveals the presence of both the BCC (ferrite/martensite) and FCC (retained austenite) phases in the weld. The amount of austenite phase is much smaller in welds obtained without beam oscillation. The amount of austenite phase decreases with a decrease in oscillation diameter from 2 mm to 0.5 mm. i.e., Rietveld analysis confirmed that the amount of retained austenite in the weld was highest (9.35%) for a beam oscillation diameter of 2 mm and was reduced (3.27%) for an oscillation diameter of 0.5 mm. It further reduced to 0.36% for welding without beam oscillation, maintaining the speed at 1000 mm/min and voltage at 60 kV constant for all welding conditions. Beam oscillation during electron beam welding brings in churning action and heat mixing in the weld pool, which is likely to reduce the thermal shear stress and shear-induced phase transformation (austenite to martensite). Therefore, more retained austenite is observed in the weld seam obtained by beam oscillation, especially for large oscillation diameters.

In the present study, the residual stresses of fusion zones were calculated by analyzing XRD data [32]. Williamson–Hall method (W–H method) was also used to estimate lattice strain and dislocation density [33,34]. Table 4 presents the residual stresses, lattice strain, and dislocation density for various welding conditions. This table also shows the hardness values in the weld and size of nonmetallic inclusions, discussed subsequently. The residual stresses were compressive irrespective of whether they were welded with beam oscillation or without beam oscillation. A similar trend in the fusion zone (i.e., compressive residual stresses) was also reported by Singh et al. for niobium weld [35] and by Ramana et al. for maraging steel [36]. The residual stress was found to increase with increasing weld speed without beam oscillation, which may be attributed to a higher cooling rate and thermal stress generation at a lower heat input with higher welding velocity. The residual stress was also found to be higher with beam oscillation than its withoutbeam oscillation case counterpart, again attributed to a higher beam speed for an oscillating beam. 

Lattice strain was also found to be higher for oscillating beams and increases with oscillation diameter from 0.5 mm to 2 mm. An increase in lattice strains may also be corroborated by the increase in dislocation density in the fusion zone, as observed in Table 4.

In the case of without beam oscillation, with decreasing speed (1200 mm/min to 1000 mm/min to 800 mm/min), the lower lattice strain may be attributed to a lower cooling rate resulting in lower thermal stress, and allowing diffusion of solute from the supersaturated matrix and its redistribution [37].

### 3.3. Mechanical Properties

#### 3.3.1. Hardness Measurements 

Figure 9a presents the hardness data of EBW joints for five welding conditions. Figure 9b presents the eleven positions from which hardness data is taken in a welded sample. Vickers hardness values showed the highest hardness in the fusion zone (FZ) and the lowest hardness in base metal. 

The fusion zone possessed the highest hardness value for all welding conditions, whether it was with beam oscillation or without beam oscillation. Additionally, HAZ possessed a comparatively lower hardness than the fusion zone for all welding conditions. However, significant hardness values were recorded both on the FZ and HAZ when compared to the base metal, similar results were recorded in the research work of Isaac et al. [38]. The hardness of the weld with beam oscillation at the highest oscillation diameter was the highest but comparable to the hardness value for the withoutbeam oscillation case case at the highest welding speed, due to the high cooling rate and fine grain structure in both cases (see Table 4). With beam oscillation, hardness increased with an increase in oscillation diameters, while in withoutbeam oscillation case cases, hardness increased with increasing weld speed. Hardness values in the fusion zone are in agreement with the values reported by F. Souza Neto et al. on the TIG and laser welding processes of medium carbon low Ni steel, AISI 4140 steel [39]. 

#### 3.3.2. Tensile Test

All weld tensile specimens produced with or without beam oscillation failed in the base metal indicating a stronger weld as presented in Figure 10. However, there was no significant difference in tensile strength between the base metal and welded specimen. The percentage of elongation was maximum in the base metal (18%), followed by the weld with beam oscillation at a 2-mm oscillation diameter (16.8%). A similar trend of elongation was observed in F. Souza Neto et al.’s research on 4140 steel welding, a medium carbon, low Ni alloy steel [39]. 

### 3.4. Wear Test

Figure 11 represents reciprocating wear test SEM images. Base metal shows severely worn debris and the weld prepared with beam oscillation at an oscillation diameter of 2 mm showed the minimum wear debris. 

Figure 12a represents the variation of the coefficient of friction (COF) as a function of interaction time. The COF is defined as the ratio of frictional force to normal force, where frictional force may depend on the asperities of different hardnesses and scales on the surface [40]. The greater the COF, the more it will be worn and the less it will be wear-resistant. COF was highest for the base metal in the range between 0.65 and 0.7. For the cases of beam oscillation, COF decreased with a decreasing oscillation diameter. 

However, COF is found to be the lowest without beam oscillation at the lowest scan speed, which is contradictory to the corresponding minimum hardness value (as seen in Table 4). This may be attributed to defects formation associated with high heat input, such as thermal stress-generated cracks, keyhole porosity and undercuts. During wear, such defective structures may show a low apparent material loss. The wear depth vs. time plot presented in Figure 12b also shows the same trend. Wear depth was highest for base metal, followed by welding with beam oscillation and then without beam oscillation. Therefore, the wear rate or wear volume is high for base metal (Figure 12c,d), but after welding, this rate is reduced. Thus, the material becomes wear resistant after welding. Similar trends were reported by Sumit et al. [41,42]. 

### 3.5. Study of Non-Metallic Inclusions (NMIs) Size Change with Welding Conditions

Figure 13 shows the morphological change of nonmetallic inclusion size for different welding conditions. Figure 13a,b represent inclusions in the weld prepared with beam oscillations at oscillation diameters of 2 mm and 0.5 mm, respectively. Figure 13c–e represent the welds without beam oscillation for varying speeds namely, 800 mm/min, 1000 mm/min and 1200 mm/min, respectively, keeping voltage and current constant. The average size of NMIs was found to be lowest at a 2-mm oscillation diameter weld condition, which is attributed to the lowest heat input and growth of inclusion. The montage image represented in Figure 13a marked by the arrow of six different locations shows a similar trend. Similarly, a montage image is provided for 0.5 mm oscillation diameter and here one can observe NMIs size variation, showing inclusion growth in some cases due to comparatively lower cooling rates. In all other cases, inclusion sizes are bigger due to comparatively higher heat input and lower cooling rates. The clear size distribution of NMIs is represented in Table 4 for different welding conditions. 

## 4. Conclusions

Welding with beam oscillation produces a churning action in the weld seam, causing heat mixing and promoting equiaxed grain and a random texture with improved ductility, hardness and wear properties. It destroys the unidirectional growth of columnar grains and the nonuniform strength distribution. Beam oscillation also enhances the beam speed and decreases the cooling rate significantly. This retards the growth of inclusions, which evolve into finer inclusions in the weld structure that are less harmful to weld. The present study with beam oscillation demonstrated these effects on the electron beam welded EN25 steel. 

Some salient conclusions that emerged out of this study are: (i)The calculated heat input rate is found to be extremely low for beam oscillation at a 2-mm oscillation diameter (6 × 10^−3^ kJ/cm) and the highest (1.57 kJ/mm) for the withoutbeam oscillation case case at 800 mm/min scan velocity.(ii)A large region of equiaxed grains was observed at the center region of the weld prepared with beam oscillation at a 2-mm oscillation diameter, attributed to churning action and heat mixing in the weld seam.(iii)The fraction of retained austenite (9.35%) was found to be highest in the weld prepared with an oscillating beam at the highest oscillating diameter of 2 mm, which was attributed to heat mixing, a lesser temperature gradient, and thermal stress and stress-induced transformation, such as austenite to martensite. Subsequently, it decreased to 3.27% with decreasing beam oscillation diameter to 0.5 mm. For withoutbeam oscillation case electron beam welding, the fraction of retained austenite was further lowered (0.36%).(iv)Residual stresses in the weld were found to be compressive in the fusion zone, irrespective of welding conditions.(v)Nonmetallic inclusion size decreased significantly for welds prepared with beam oscillation, especially at higher oscillating diameters, which was attributed to the fastest cooling rate that retarded the growth of inclusion.(vi)The hardness and wear properties of welds were found to improve after welding, especially for welds with oscillating beams.

### 4.1. Limitation of the Present Work

In the present work, the welding speed and beam oscillation diameter have been studied, whereas the welding current and frequency have been kept constant. 

Additionally, to achieve full-depth penetration of all the samples, the optimum welding current with a 2-mm beam oscillation diameter has been considered for carrying out welding for other cases. This makes the apparent current input higher than required for other cases to achieve full penetration. 

### 4.2. Future Scope

A tailored welded blank of MCLA with stainless steel (SS) has found widespread application in the construction process in production industries [43]. Therefore, dissimilar joining of EN25 steel to SS by EBW could be explored.

Effects of frequency and beam oscillation pattern could also be explored.

## Figures and Tables

**Figure 1 materials-16-02717-f001:**
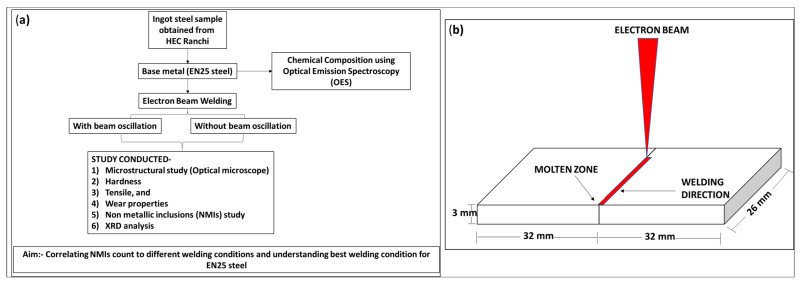
Workflow of the present study, (**a**) methodology and (**b**) electron beam welding on EN25 base material, with/without beam oscillation.

**Figure 2 materials-16-02717-f002:**
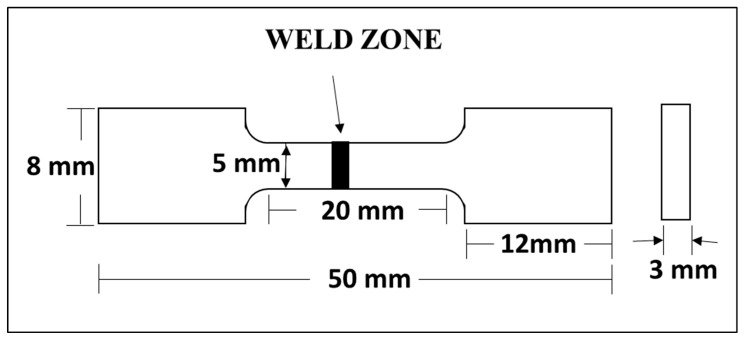
Schematic of tensile specimen geometry used after electron beam welding.

**Figure 3 materials-16-02717-f003:**
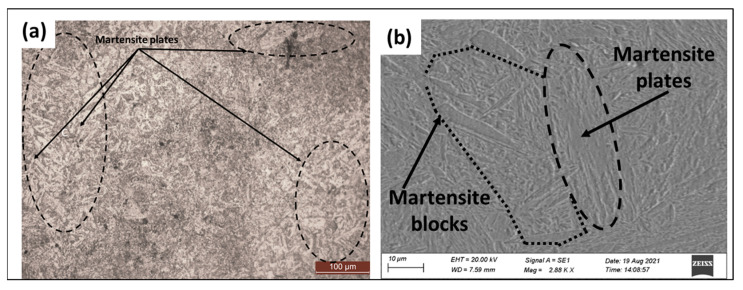
Microstructure of base material (EN25 Steel) (**a**) optical micrograph, (**b**) scanning electron micrograph.

**Figure 4 materials-16-02717-f004:**
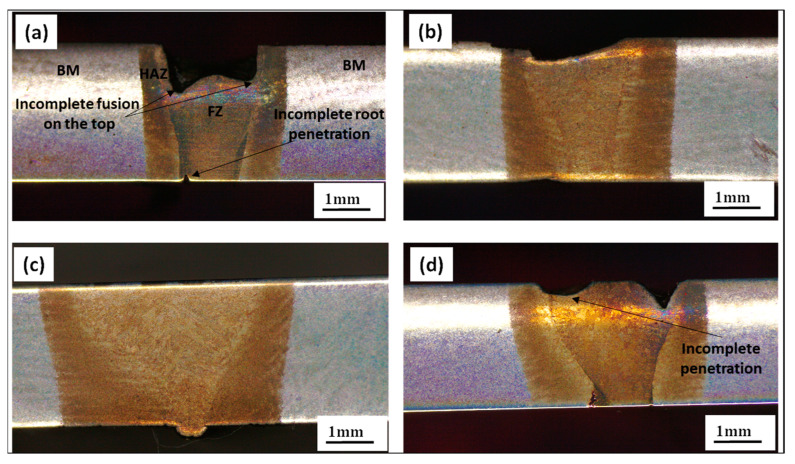
The cross-sectional appearance of weld bead under different weld conditions (**a**) S1000V60I35 OD 2 mm, (**b**) S1000V60I35 OD 0.5 mm, (**c**) S1000V60I35 WO and (**d**) S1200V60I35 WO.

**Figure 5 materials-16-02717-f005:**
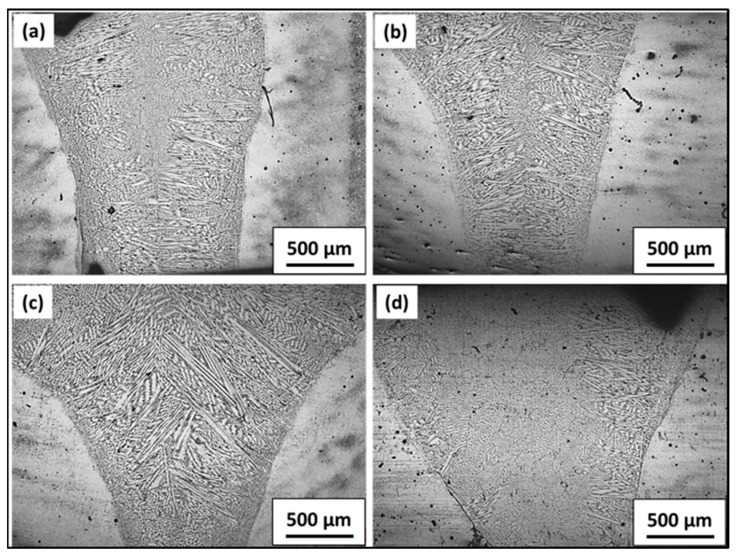
Optical images for different welding conditions at 20× magnification and 5× magnification at the top right corner. (**a**) S1000V60I35 OD 2 mm, (**b**) S1000V60I35 OD 0.5 mm, (**c**) S1000V60I35 WO and (**d**) S1200V60I35 WO.

**Figure 6 materials-16-02717-f006:**
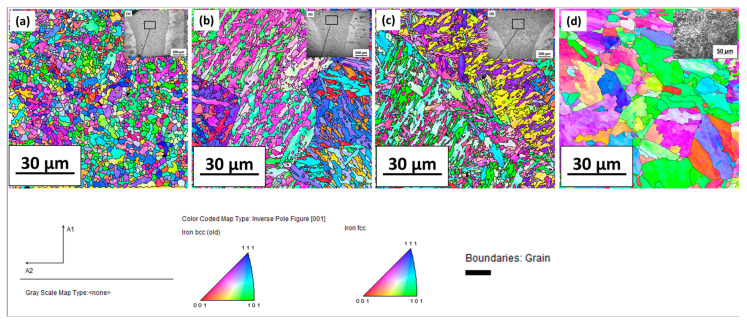
IPF maps with and without beam oscillation conditions for (**a**) S1000V60I35 OD2 mm, (**b**) S1000V60I35 OD 0.5 mm, (**c**) S1000V60I35 WO and (**d**) base metal.

**Figure 7 materials-16-02717-f007:**
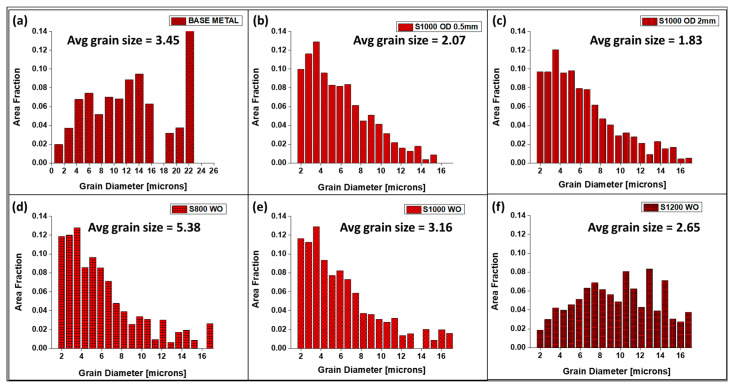
Grain size distribution for different sample conditions: (**a**) base metal, (**b**) S1000V60I35 OD 0.5 mm, (**c**) S1000V60I35 OD 2 mm (**d**) S800V60I35 WO, (**e**) S1000V60I35 WO and (**f**) S1200V60I35.

**Figure 8 materials-16-02717-f008:**
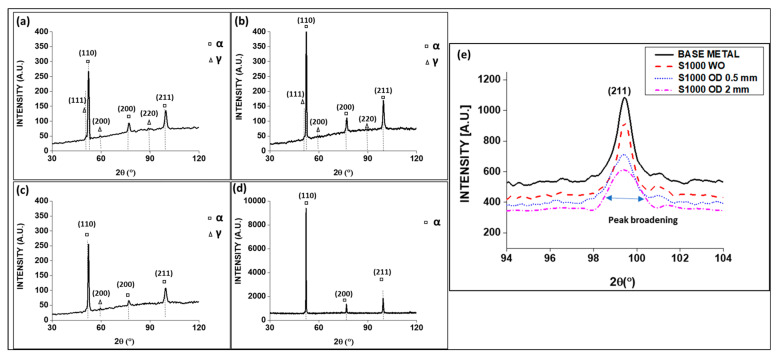
XRD peak for (**a**) S1000V60I35 OD 2 mm, (**b**) S1000V60I35 OD 0.5 mm, (**c**) S1000V60I35 WO, (**d**) base metal and (**e**) microstrain calculation from peak broadening for all sample conditions.

**Figure 9 materials-16-02717-f009:**
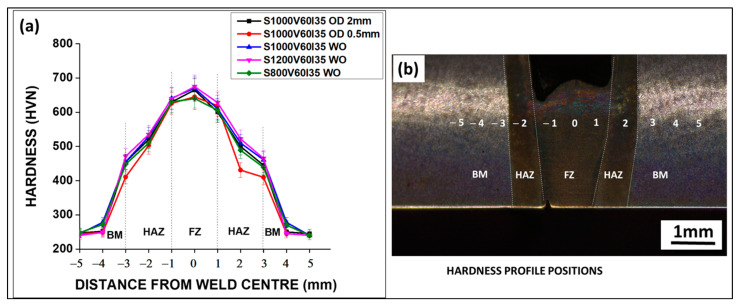
(**a**) Hardness measurement of EN25 steel after welding under different welding conditions, from base metal to heat-affected zone to fusion zone, (**b**) EN25 welded steel sample at different positions from where hardness measurement is taken, starting from point −5 to 5 in (**a**).

**Figure 10 materials-16-02717-f010:**
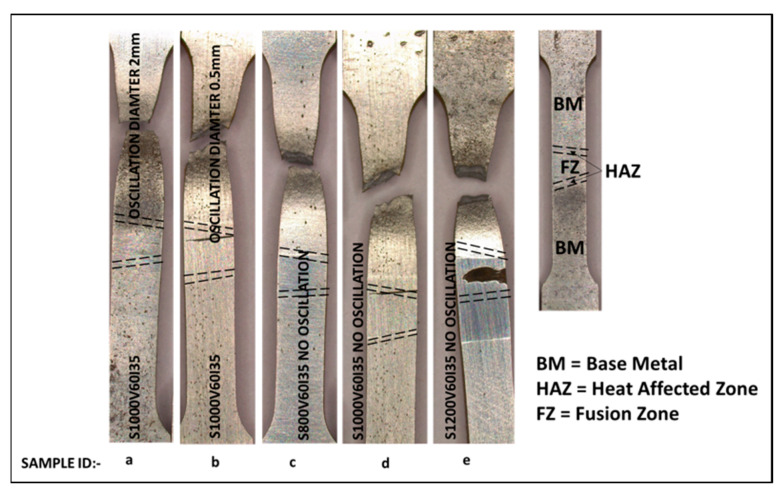
Tensile sample after the test for different sample conditions (**a**) S1000V60I35 OD 2 mm, (**b**) S1000V60I35 OD 0.5 mm (**c**) S800V60I35 WO, (**d**) S1000V60I35 WO and (**e**) S1200V60I35.

**Figure 11 materials-16-02717-f011:**
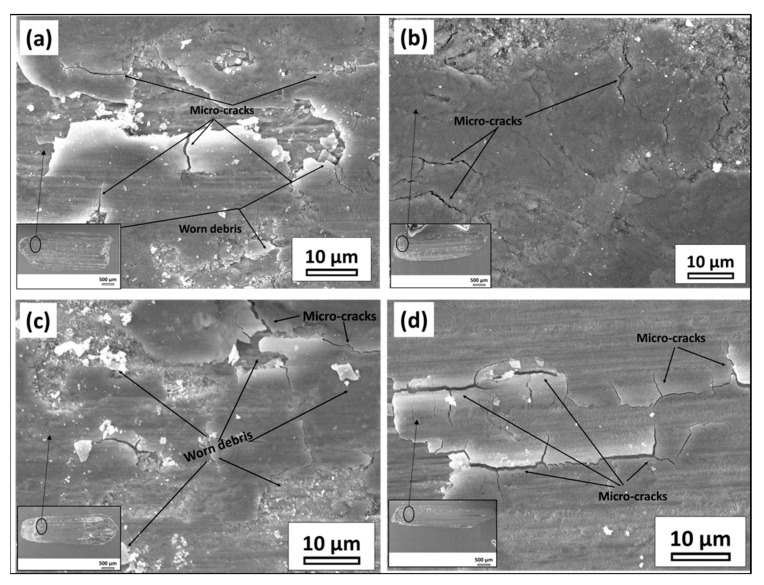
Wear test of EN25 steel for different sample conditions (**a**) Base metal, (**b**) S1000V60I35 OD 2 mm, (**c**) S1000V60I35 OD 0.5 mm, and (**d**) S1000V60I35 WO, all for load 30 N track length 2 mm and frequency 10 Hz.

**Figure 12 materials-16-02717-f012:**
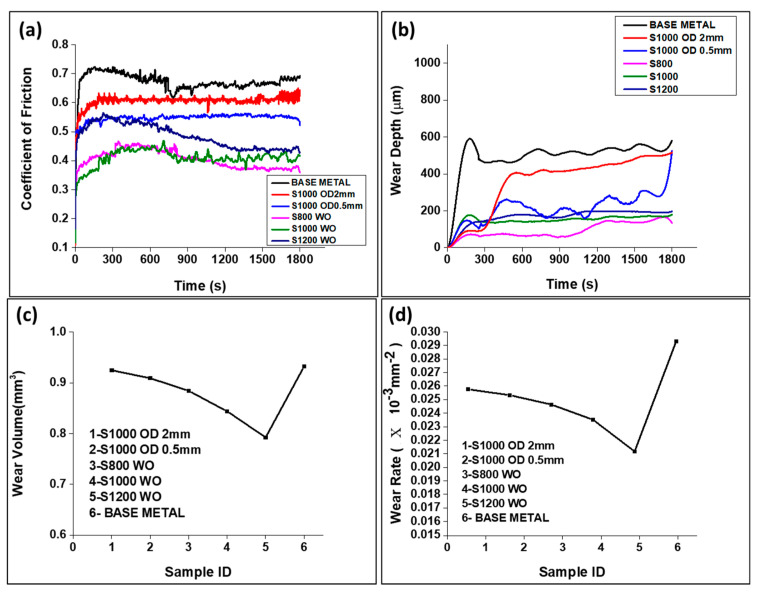
Wear test of EN25 steel for different sample conditions (**a**) COF vs. time, (**b**) wear depth vs. time, (**c**) wear volume vs. time and (**d**) wear rate vs. time, all for load 30 N track length 2 mm and frequency 10 Hz.

**Figure 13 materials-16-02717-f013:**
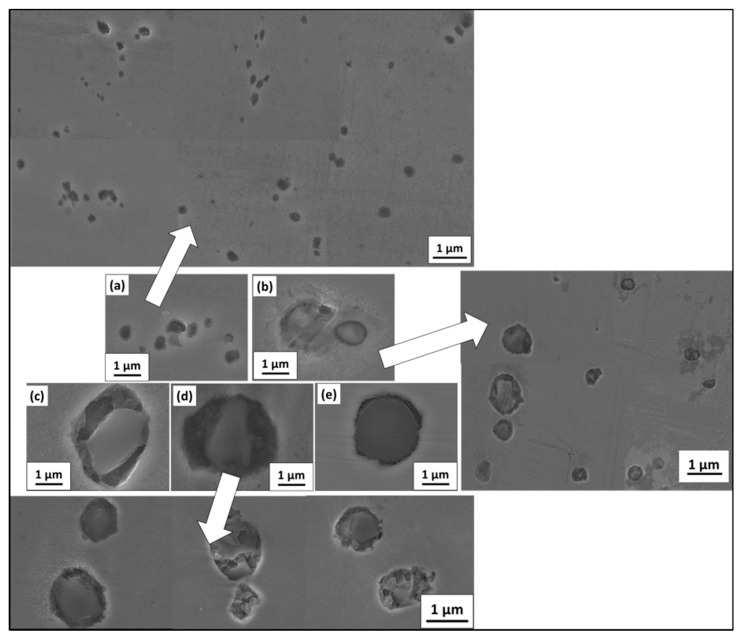
Nonmetallic inclusion size variation under different weld conditions (**a**) S1000V60I35 OD 2 mm, (**b**) S1000V60I35 OD 0.5 mm, (**c**) S800V60I35 WO, (**d**) S1000V60I35 WO and (**e**) S1200V60I35 WO.

**Table 1 materials-16-02717-t001:** Chemical composition of the investigated EN25 steel in wt%.

C	Si	Mn	Ni	Cr	Mo	S	P	Fe
0.32	0.28	0.56	2.4	0.72	0.62	0.02	0.025	95.055

**Table 2 materials-16-02717-t002:** Parameter of electron beam welding with full penetration at 3 mm depth for all welding conditions.

S.No.	Sample ID	Speed (S) (mm/min)	Voltage (V) (kV)	Current (I) (mA)	With Beam Oscillation		Without Beam Oscillation (WO)
					Oscillation Diameter (OD) (mm)	Oscillation Frequency (Hz)	
1.	S1000V60I35 OD 2 mm	1000	60	35	2	300	No
2.	S1000V60I35 OD 0.5 mm	1000	60	35	0.5	300	No
3.	S800V60I35 WO	800	60	35	-	-	Yes
4.	S1000V60I35 WO	1000	60	35	-	-	Yes
5.	S1200V60I35 WO	1200	60	35	-	-	Yes

**Table 3 materials-16-02717-t003:** Measured weld bead and calculated heat input comparisons for all five conditions.

Serial No.	Sample ID	Measured Weld Bead (FZ) (mm)	Calculated Heat Input (Q) (kJ/cm)
1.	S1000V60I35 OD 2 mm	2.077	0.006
2.	S1000V60I35 OD 0.5 mm	2.617	0.024
3.	S800V60I35 WO	2.924	1.57
4.	S1000V60I35 WO	2.813	1.26
5.	S1200V60I35 WO	2.626	1.05

**Table 4 materials-16-02717-t004:** Residual stress, lattice strain and dislocation density, hardness, wear rate and NMIs size calculation of fusion zone for different sample conditions.

Serial No.	Sample ID	Lattice Strain %	Residual Stress (MPa)	Dislocation Density (mm^−2^)	Hardness of FZ (HVN)	Wear Rate (10^−3^) (mm^−2^)	NMIs Size Range (µm)
1.	S1000V60I35 OD 2 mm	0.489	−414.4 ± 54	6.34 × 10^15^	665 ± 5	0.02576	0.1–0.52
2.	S1000V60I35 OD 0.5 mm	0.341	−300 ± 40	3.19 × 10^15^	645 ± 5	0.02533	0.56–1.78
3.	S800V60I35 WO	0.115	−288.59 ± 98	3.98 × 10^14^	640 ± 5	0.02363	1–5
4.	S1000V60I35 WO	0.124	−215.4 ± 13	5.44 × 10^14^	670 ± 5	0.02351	1–5
5.	S1200V60I35 WO	0.236	−198.6 ± 88	9.82 × 10^14^	675 ± 5	0.02117	1–3
6.	BASE METAL	0.096	−150.71 ± 19	1.66 × 10^14^	240 ± 5	0.02931	5–10

## Data Availability

Not applicable.

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
