# Peer review of "Effect of Beam Oscillation on Microstructure and Mechanical Properties of Electron Beam Welded EN25 Steel"

_materials, 2023, doi:10.3390/ma16072717_

Round 1

Reviewer 1 Report

The topic is interesting also all required analysis were desribed in deep. I have no comments. The paper could be published.

Author Response

Response 1: On behalf of all authors, I thank the reviewer for his positive comments about our research work and manuscript.

Reviewer 2 Report

From the analysis of literature sources, the use of beam vibrations has led to an improvement in the quality of the weld of dissimilar materials (grain size decreases, grain equiaxity, uniformity of microstructure, reduction of inclusions, increased toughness, high tensile strength, etc.). The authors claim that the reason for improving the quality of the weld is thermal mixing and the formation of a random texture.

I didn't see any proof of this statement, so in the review I pointed to a link that provides computer modeling and analysis of solidification processes depending on the time of action of the heat flow. One of the co-authors of the article (G. G. Roy) is a specialist in welding modeling, and in my opinion, it will not be difficult for him to explain the statement made, and then the article will significantly improve and become a methodological basis for periodically changing heat flow during welding.

The experimental part of the article is carried out at a high scientific level and is of interest to welders.

The answers to the questions are marked in the tables.

Page 22-23: Beam oscillation resulted in equiaxed grain in the center region of the fusion zone, attributed to the heat mixing and evolution of random texture.

This statement requires proof, for example, using numerical modeling

[Ziyou Yang; Jingshan He; (2021)].

Ziyou Yang; Jingshan He; (2021). Numerical investigation on fluid transport  phenomena in electron beam welding of aluminum alloy: Effect of the focus position and incident beam angle on the molten pool behavior . International Journal of Thermal Sciences, (), –. doi:10.1016/j.ijthermalsci.2021.106914

Author Response

Point1: From the analysis of literature sources, the use of beam vibrations has led to an improvement in the quality of the weld of dissimilar materials (grain size decreases, grain equiaxity, uniformity of microstructure, reduction of inclusions, increased toughness, high tensile strength, etc.). The authors claim that the reason for improving the quality of the weld is thermal mixing and the formation of a random texture.

I didn't see any proof of this statement, so in the review I pointed to a link that provides computer modeling and analysis of solidification processes depending on the time of action of the heat flow. One of the co-authors of the article (G. G. Roy) is a specialist in welding modeling, and in my opinion, it will not be difficult for him to explain the statement made, and then the article will significantly improve and become a methodological basis for periodically changing heat flow during welding.

The experimental part of the article is carried out at a high scientific level and is of interest to welders.

The answers to the questions are marked in the tables.

Response 1: We thank the reviewer for his excellent suggestions and as suggested Prof G G Roy has made significant work in in this area and we cite some of those works relevant to this study here. We also thank the reviewer for appreciating our experimental work.

Point 2: Page 22-23: Beam oscillation resulted in equiaxed grain in the center region of the fusion zone, attributed to the heat mixing and evolution of random texture.

This statement requires proof, for example, using numerical modeling

[Ziyou Yang; Jingshan He; (2021)].

Ziyou Yang; Jingshan He; (2021). Numerical investigation on fluid transport phenomena in electron beam welding of aluminum alloy: Effect of the focus position and incident beam angle on the molten pool behavior . International Journal of Thermal Sciences, (), –. doi:10.1016/j.ijthermalsci.2021.106914.

Response 2:  We thank reviewer for his comments and as suggested we included the above mentioned paper in the references.

Reviewer 3 Report

1.      In line 350, the authors have been initialled coefficient of friction with “COF”, but in line 355 the authors still state it with coefficient of friction, please be consistent. If want use abbreviation of COF, please use it along the manuscript after mentioning first time coefficient of friction.

2.      The authors needs to explain in brief of Coefficient of friction. According to Jamari et al., The asperity of contact interface between two bodies was defined by the coefficient of friction. This value was obtained from an experimental setup, either pin-on-disc. The authors encouraged to include this explanation. Relevant reference needs to incorporated as follow: Computational Contact Pressure Prediction of CoCrMo, SS 316L and Ti6Al4V Femoral Head against UHMWPE Acetabular Cup under Gait Cycle. J. Funct. Biomater. 2022, 13, 64. https://doi.org/10.3390/jfb13020064

3.      Additional figure in introduction would be improve the presentation of manuscript and also giving better understanding to the reader.

4.      Describe the novelty of the article made by the author? From the results of my evaluation, it seems that many similar published works adequately explain what you have raised in the current manuscript. If there is something others really new in this manuscript, please highlight it more clearly in the introduction section.

5.      Previous research has to be explained in the introduction section, including their work, novelty, and limits, to illustrate the research gaps that will be filled in the current study.

6.      In the materials and methods, the authors need to add additional illustrations as a form of figure that explains the workflow of the present study to make the reader easier to understand rather than only the dominant text as a present form.

7.      More detail in tools information such as manufacturer, country, and specification need to be stated.

8.      The abstract section should include quantitative results.

9.      Please add the abstract's "take-home" message, the current form was insufficient.

10.   Rearrange keywords alphabetically.

Author Response

Point1: In line 350, the authors have been initialled coefficient of friction with “COF”, but in line 355 the authors still state it with coefficient of friction, please be consistent. If want use abbreviation of COF, please use it along the manuscript after mentioning first time coefficient of friction.

Response 1: Thank you for your suggestions and corrections and we now taken care of it.

Point 2: The authors needs to explain in brief of Coefficient of friction. According to Jamari et al., The asperity of contact interface between two bodies was defined by the coefficient of friction. This value was obtained from an experimental setup, either pin-on-disc. The authors encouraged to include this explanation. Relevant reference needs to incorporated as follow: Computational Contact Pressure Prediction of CoCrMo, SS 316L and Ti6Al4V Femoral Head against UHMWPE Acetabular Cup under Gait Cycle. J. Funct. Biomater. 2022, 13, 64. https://doi.org/10.3390/jfb13020064

Response 2: We thanks reviewer for his suggestions and as suggested we cited relevance reference.

Point3: Additional figure in introduction would be improve the presentation of manuscript and also giving better understanding to the reader.

Response 3: Thank you for suggestion and now we included fig 1 showing the work flow, EBS set up etc.

Point 4: Describe the novelty of the article made by the author? From the results of my evaluation, it seems that many similar published works adequately explain what you have raised in the current manuscript. If there is something others really new in this manuscript, please highlight it more clearly in the introduction section.

Response 4: It is evident from previous research studies that lot has been studied on effect of weld parameters on microstructure and mechanical properties of steels. However, in this study, we used EBW technique with different beam oscillation diameters to understand the changes in inclusions and microstructure formation in EN steel grade, which was not extensively studies in the past.  The last paragraph of the introduction section has been modified. 

Point 5: Previous research has to be explained in the introduction section, including their work, novelty, and limits, to illustrate the research gaps that will be filled in the current study.

Response 5: As suggested, we have now explained previous studies and novelty in our work in the last paragraph of the introduction section of this article. 

Point 6: In the materials and methods, the authors need to add additional illustrations as a form of figure that explains the workflow of the present study to make the reader easier to understand rather than only the dominant text as a present form.

Response 6: Thank you for your suggestions I have added workflow in figure 1 for clear understanding.

Point 7: More detail in tools information such as manufacturer, country, and specification need to be stated.

Response 7: Sufficient information has been provided related to base metal and from where it was acquired.

Point 8: The abstract section should include quantitative results.

Response 8: Thank you for the suggestion and the abstract had already discussed about the quantitative results in terms of effect of beam oscillation diameters on the fraction of austenite and the average size of non-metallic inclusions (line #15 to 23) and now we included hardness and wear resistance quantified results.

Point 9: Please add the abstract's "take-home" message, the current form was insufficient.

Response 9: I think “take-home” message is included in the abstract. 

Point 10: Rearrange keywords alphabetically.

Response 10: Thank you for your kind suggestion and we now rearranged keywords as suggested.

Reviewer 4 Report

The submitted manuscript deals with an experimental investigation on effect of beam oscillation on microstructure and mechanical properties of electron beam welded EN25 steel joints. The analyzed topic is well included between the main scope of the SI.

Referee is, in general, positive toward this submission, nevertheless some drawbacks deserve further attention, as detailed in what follows:

Some references are wrongly referenced in the text. For example, paper 12 should be cited as "Matsuda and Ueyama" rather than "Ueyama et al."

line 104: "CSLM"? All acronyms should be defined in the text.

Figures and Tables should be placed after being referenced in the text.

Table 1. Chemical composition measurements were not performed correctly. The sum of the elements in the table is about 5wt.%. So what's the rest?

The research plan is too modest. Constant voltage, current, and only three speeds were studied. Moreover, in the case of the WO specimens the welding test have been carried for only two oscillation diameters, keeping the rest of parameters constant. This is too little to obtain reliable and far-reaching conclusions. It is a preliminary study that requires an extension of the research plan. In addition, the choice of parameter values was not justified.

Figure 5. Why are the results not shown for the S800V60I35WO?

Figure 6: Why are the results not shown for the S800V60I35WO and S1200V60I35WO?

Equations presented in the manuscript are well known. Their sources should be cited.

Figure 8 should be enlarged. The font in the drawings is also too small.

line 289: "W-H method" ?

line 317: "Isaac et al. [33]" ?

section 3.3.2: Tensile curves should be presented in the manuscript.

Author Response

Point 1: Table 1. Chemical composition measurements were not performed correctly. The sum of the elements in the table is about 5wt.%. So what's the rest?

Response 1: Thank you for the suggestion.  Chemical composition measurement has been done correctly. I missed to add the percentage of iron, which is the major amount,  in the Table-1.  Now it is added.

Point2: The research plan is too modest. Constant voltage, current, and only three speeds were studied. Moreover, in the case of the WO specimens the welding test have been carried for only two oscillation diameters, keeping the rest of parameters constant. This is too little to obtain reliable and far-reaching conclusions. It is a preliminary study that requires an extension of the research plan. In addition, the choice of parameter values was not justified.

Response 2: It is evident from previous research studies that a lot has been studied on the effect of weld parameters on microstructure and mechanical properties of steels. However, in this study, we used EBW technique with different beam oscillation diameters to understand the changes in inclusions and microstructure formation in EN steel grade, which was not extensively studies in the past.  Even with two oscillation diameters very contrasting results have emerged, which would help the future researchers to follow. 

Point 3: Figure 5. Why are the results not shown for the S800V60I35WO?

Response 3: Thank you for pointing out. In case of S800V60I35WO, because of very high heat input welding defect occurred and we discarded the image. 

Point 4: Figure 6: Why are the results not shown for the S800V60I35WO and S1200V60I35WO?

Response 4: Thank you for putting up this query. Here we wanted to compare the grain orientation comparing the beam oscillation to its non-oscillating counterpart at a constant linear welding speed, 1000mm/min.  Therefore, other speeds in case of non-oscillating case were not considered. 

Point 5: Section 3.3.2: Tensile curves should be presented in the manuscript.

Response 5: Thank you for your kind suggestion. Since we carried out the tensile experiments with miniature samples, somehow tensile curve could not be generated properly.  But since all weld failed in base mental, it indicated that weld were stronger than base metal.  In the revised manuscript, I have added Fig.10 showing tensile samples failing in the base metal only. 

Point 6: Equations presented in the manuscript are well known. Their sources should be cited.

Response 6: Thank you for notifying this I have incorporated the source from where I have taken the equation.

Point 7: Figure 8 should be enlarged. The font in the drawings is also too small.

Response 7: Thank you for your kind suggestion. I have changed the font size now the figure is visible very clearly.

Point 8: line 289: "W-H method" ?

Response 8: thank you for notifying. I have made correction and written the full form of the technique by which lattice strain and dislocation density is calculated more accurately.

Point 9: line 317: "Isaac et al. [33]" ?

Response 9: Thank you for your kind suggestion. Sentence was not placed correctly previously this time I have corrected the sentence this time.

Round 2

Reviewer 2 Report

After the additions have been made, the article can be published.

Author Response

Point 1: After the additions have been made, the article can be published.

Response 1:  Thank you for accepting our manuscript.

Reviewer 3 Report

1.      Please do not use abbreviations in keywords.

2.      In line 103, the authors mention first time of “SEM”, please explain the stand of its abbreviation of SEM.

3.      Why the present study performs only experimental testing? Why not combined with analytical and/or computational study? It would improve the present article in broader scientific contribution.

4.      In line 359-364, incorporated others relevant reference related to coefficient of friction as follows, doi: 10.3390/fluids7070225, 10.3390/su142013413, and 10.3390/jfb12020038

5.      Error and tolerance of experimental tools used in this work are important information that needs to be explained in the manuscript. It is would use as a valuable discussion due to different results in the further study by other researcher.

6.      Findings must be compared to similar past research.

7.      The discussion in this article is too poor. Significant improvement is required, particularly when discussing the results rather than just mentioning the results with a brief explanation.

8.      Please include the limitation of the present study, it is missing.

9.      In the conclusion, please explain the further research.

10.   Five years back literature should be enriched into the reference. MDPI reference is strongly recommended.

11.   In the whole of the manuscript, the authors sometimes made a paragraph only consisting of one or two sentences that made the explanation not clearly understood. The authors need to extend their explanation to become a more comprehensive paragraph. In one paragraph, it is recommended to consist of at least 3 sentences with 1 sentence as the main sentence and the other sentences as supporting sentences.

12.   Provide graphical abstract for submission after revision.

13.   The manuscript needs to be proofread by the authors since it has grammatical and language issues.

Author Response

Point 1:  Please do not use abbreviations in keywords.

Response 1: Thank you for the suggestion. Except for the steel grade name, no abbreviation is used in keywords.

Point 2: In line 103, the authors mention the first time of “SEM”, please explain the stand of its abbreviation of SEM.

Response 2: Thank you for the suggestion. I have provided the full form of SEM at the first use in the text (line 113 in the revised-2 manuscript).

Point 3: Why the present study performs only experimental testing? Why not combined with analytical and/or computational study? It would improve the present article in broader scientific contribution.

Response 3: We agree analytical or computational results help in understanding the phenomena.  However, it is not mandatory for the present article to have such results.  It is purely an experimental work and results have been discussed with reference to available literature and interpreted.  The article presents some meaningful results, which will help the researcher to understand the weldability of EN25 steel with circular beam oscillation at different oscillation diameters. All the conclusions have been supported by the experimental results. 

Point 4: In line 359-364, incorporated others relevant reference related to coefficient of friction as follows,

1) doi: 10.3390/fluids7070225, “Performance Comparison of Newtonian and Non-Newtonian Fluid on a Heterogeneous Slip/No-Slip Journal Bearing System Based on CFD-FSI Method”, Fluids,

2) doi: 10.3390/su142013413, “Minimizing Risk of Failure from Ceramic-on-Ceramic Total Hip Prosthesis by Selecting Ceramic Materials Based on Tresca Stress”, Sustainability, and

3) doi: 10.3390/jfb12020038,  “The Effect of Bottom Profile Dimples on the Femoral Head on Wear in Metal-on-Metal Total Hip Arthroplasty”, J. Functional. Biomaterials

Response 4: Welcome suggestion. I think these references will be superfluous only to support the definition of COF in the manuscript.  Already one reference has been cited.

Point 5: Error and tolerance of experimental tools used in this work are important information that needs to be explained in the manuscript. It is would use as a valuable discussion due to different results in further study by other researchers.

Response 5: Thank you for the comments. I have added the hardness, tensile and wear instrument details.  I have also discussed results with reference to the published literature, whenever possible.

Point 6: Findings must be compared to similar past research.

Response 6: Thank you for your suggestion.  We have added a few relevant literature for comparing with past results, whenever applicable. 

Point 7: The discussion in this article is too poor. Significant improvement is required, particularly when discussing the results rather than just mentioning the results with a brief explanation.

Response 7: Thank you for your kind suggestion. I have further modified the discussion of results, and cited previous literature, wherever possible.

Point 8: Please include the limitation of the present study, it is missing.

Response 8:  Thank you for your kind suggestion. I have added the limitation of the present work in conclusion section. 

Point 9: In the conclusion, please explain further research.

Response 9: I have further added the future scope of work in the conclusion section.

Point 10: Five years back literature should be enriched into the reference. MDPI reference is strongly recommended.

Response 10: Thank you for your suggestion. I have added literature that supports the necessary results. We have already added 2 MDPI journals.  Unfortunately, we could not find much MDPI papers relevant to us.  Our 60% references are after 2018 only, i.e., in the last 5 years. 

Point 11: In the whole of the manuscript, the authors sometimes made a paragraph only consisting of one or two sentences that made the explanation not clearly understood. The authors need to extend their explanation to become a more comprehensive paragraph. In one paragraph, it is recommended to consist of at least 3 sentences with 1 sentence as the main sentence and the other sentences as supporting sentences.

Response 11: We have enhanced the discussion and cited literature, whenever possible. 

Point 12: Provide graphical abstract for submission after revision.

Response 12: Thank you for your suggestion we have uploaded a graphical abstract. 

Point 13: The manuscript needs to be proofread by the authors since it has grammatical and language issues.

Response 13: I have checked and corrected the grammatical error in the manuscript.

Reviewer 4 Report

I would like to thank the authors for the effort they have made in replying to all my comments and concerns.
I am satisfied with the reviewed version and the provided explanations to all raised questions.

Author Response

Point 1: I would like to thank the authors for the effort they have made in replying to all my comments and concerns. I am satisfied with the reviewed version and the provided explanations to all raised questions.

Response 1:  Many thanks for considering our response and accepting the manuscript.